# Ga-68-PSMA-11 PET/CT in Patients with Biochemical Recurrence of Prostate Cancer after Primary Treatment with Curative Intent—Impact of Delayed Imaging

**DOI:** 10.3390/jcm11123311

**Published:** 2022-06-09

**Authors:** Jolanta Kunikowska, Kacper Pełka, Omar Tayara, Leszek Królicki

**Affiliations:** 1Nuclear Medicine Department, Medical University of Warsaw, 02-091 Warsaw, Poland; jolanta.kunikowska@wum.edu.pl (J.K.); leszek.krolicki@wum.edu.pl (L.K.); 2Department of Methodology, Laboratory of Center for Preclinical Research, Medical University of Warsaw, 02-091 Warsaw, Poland; 3Second Department of Urology, Centre of Postgraduate Medical Education, 01-809 Warsaw, Poland; o.tayara@wp.pl

**Keywords:** PSMA prostate cancer, delayed phase, PET/CT, Ga-68-PSMA-11A

## Abstract

The presence of prostate-specific membrane antigen (PSMA) on prostate cancer cells and its metastases allows its use in diagnostics using PET/CT. The aim of this study was to evaluate the usefulness of delayed phase images in the Ga-68-PSMA-11 PET/CT. **Methods:** 108 patients with prostate cancer (median age: 68.5 years, range: 49–83) were referred for Ga-68-PSMA-11 PET/CT due to biochemical relapse (PSA (prostate-specific antigen) (3.2 ± 5.4 ng/mL). Examinations were performed at 60 min, with an additional delayed phase of the pelvis region at 120–180 min. **Results:** The Ga-68-PSMA-11 PET/CT showed lesions in 86/108 (80%) patients; detection rate depending on the PSA level: 0.2 < PSA < 0.5 ng/mL vs. 0.5 ≤ PSA < 1.0 ng/mL vs. 1.0 ≤ PSA < 2.0 ng/mL vs. PSA ≥ 2.0 ng/mL was 56% (standard vs. delay: 56 vs. 56%) vs. 60% (52 vs. 60%) vs. 87% (83 vs. 87%) vs. 82% (77 vs. 82%) of patients, respectively. The delayed phase had an impact on the treatment in 14/86 patients (16%) (*p* < 0.05): 7 pts increased uptake was seen only after 60 min, which was interpreted as physiological or inflammatory accumulation; the delayed image showed increased accumulation in 7 patients only: 4 in regional lymph nodes, 1 in local recurrence, and 2 patients with local recurrence showed additional foci. **Conclusions:** Delayed phase of Ga-68-PSMA-11 PET/CT has an impact on treatment management in 16% of patients.

## 1. Introduction

Prostate cancer is the most frequently diagnosed cancer in men [1]. In many cases, the tumor is slow-growing, and most men with low-risk prostate cancer can live for many years. In the case of more aggressive tumors, the primary treatment with curative intent—a radical prostatectomy (RP) or radiotherapy—is the method of choice. Unfortunately, even with proper therapy, about 20–40% of men will have a recurrence during the five-year follow-up [2,3]. In patients undergoing follow-up, the cost-effective and approved method to detect disease relapse is monitoring the level of the prostate-specific antigen (PSA).

The biochemical recurrence (BCR) after RP has been defined as a PSA level rising above 0.2 ng/mL. However, the latest European Association of Urology (EAU) guideline suggests this threshold should be 0.4 ng/mL to best predict metastases after RP [4]. After radiation therapy, according to the American Society for Therapeutic Radiation and Oncology (ASTRO), any PSA increase 2 ng/mL greater than the nadir value is considered as BCR, regardless of the serum concentration of the nadir [5].

Computed tomography (CT), magnetic resonance imaging (MRI), and PET/CT with different radiolabeled tracers are used to find the site of the relapse. Each of them has well-known advantages and disadvantages. CT and MRI give the possibility to visualize the structure. Molecular imaging shows physiology. The important fact pointed out by EAU guidelines is that any imaging method is only justified in patients for whom the findings will affect treatment decisions [4].

Prostate-specific membrane antigen (PSMA) PET imaging is the method with the highest detection rate in cases of BCR, even in cases of low PSA levels [6,7,8]. PSMA is a transmembrane protein present on the outer surface, with expression 100–1000-fold higher in prostate cancer cells than the baseline physiologic expression in other cells. The PSMA expression increases with the tumor grade [9,10] and with the PSA level. For PSMA PET/CT imaging, several PSMA-based radiopharmaceuticals have become available, which are labeled with Ga-68 or F-18 such as Ga-68-PSMA HBED-CC (PSMA-11), Ga-68-DOTAGA-(3-iodo-y)fk(Sub-KuE) (PSMA I&T), F-18-DCFBC, F-18-DCFPyL, and the recently developed F-18-PSMA-1007.

According to the EANM and the SNMMI guidelines for isotope use, a standard procedure with a 60 min interval for uptake time is recommended for the Ga-68-PSMA-11, with an acceptable range from 50 to 100 min [11]. There is limited data in the literature regarding the usage of delayed phase imaging up to 3–4 h after injection of radioisotope [12,13,14,15,16].

We have reported the case of a patient with prostate adenocarcinoma, Gleason score 9, and BCR after treatment with curative intent when the standard Ga-68-PSMA-11 PET/CT examination, 60 min post injection, was negative. Only the delayed phase, 2 h post injection, showed the recurrence of the disease in the right internal iliac lymph node in that patient [17]. Based on that case, we developed this prospective study.

The aim of this paper was to determine the advantages of dual-point early and delayed Ga-68-PSMA-11 PET/CT in patients with BCR of prostate cancer after primary treatment with curative intent.

## 2. Materials and Methods

### 2.1. Patients’ Population

This prospective study was conducted between March 2019 and August 2020, with the screening of 154 patients. Finally, the study population consisted of 108 men (median 68.5 years; range: 49–83 years) who met the inclusion criteria (flowchart). All patients were referred for a Ga-68-PSMA-11 PET/CT due to BCR after primary treatment with curative intent for prostate cancer.

In the study, we included patients with a history of prostate cancer, verified by histopathological evaluation, and presented with BCR/persistence as determined by PSA levels.

The inclusion criteria were: patients after radical treatment (prostatectomy, radical radiotherapy, high-intensity focused ultrasound (HIFU)); histologically confirmed adenocarcinoma of the prostate, Gleason score ≥ 5; BCR: defined as PSA is ≥0.2 ng/mL in minimum two consecutive measurements; in patients after only RTx, the BCR was related to an increase in PSA level of at least 2 ng/mL, compared to the nadir; no suspicion of bone metastases in other imaging methods; prior major surgery must be at least 12 weeks prior to study entry and Eastern Cooperative Oncology Group (ECOG) performance status 0 to 2, with a life expectancy ≥ 6 months.

The exclusion criteria were: male age under 18; other known coexisting malignancies except for non-melanoma skin or low-grade superficial bladder cancer unless successfully treated and proven no evidence of recurrence for 5 years; any evidence of severe or uncontrolled systemic or psychiatric diseases, screening for chronic conditions was not required; subject with lack of cooperation; subject with lack of informed consent to participate in the study; subjects who have taken part in other clinical trials within the last 28 days were excluded from participation in this study; known allergy, hypersensitivity, or intolerance to PSMA and subject unlikely to comply with study procedures, restrictions, and requirements and judged by the investigator to be unsuitable for study participation.

All patients had previously undergone primary treatment with curative intent. Prostatectomy was performed in 84 patients, 44 of whom had additional radiotherapy. A total of 22 patients underwent only radical external beam radiation therapy (EBRT), and 2 had HIFU therapy. In the study group, the average PSA concentration at the time of the PET/CT was 3.2 ± 5.4 ng/mL (range: 0.2–39.0).

### 2.2. Ga-68-PSMA-11 PET/CT Protocol and Image Interpretation

Radiopharmaceutical preparation was performed as previously described [18]. The Ga-68-PSMA-11 was injected intravenously 60 min before the PET/CT acquisition. Patients received a dose of between 135 and 220 MBq (3.65–5.95 mCi) (2.0 ± 0.1 MBq/kg 0.05 ± 0.0003 mCi/kg) to meet the criterion of 2 MBq/kg (0.05 mCi/kg) body mass. Forced diuresis was induced with 20 mg of furosemide administered intravenously just after Ga-68-PSMA-11 injection to improve the image quality. The PET/CT was performed on a Biograph 64 TruePoint (Siemens Medical Solutions Inc., USA) from skull apex to thighs with a 3 min acquisition time per bed position (reconstruction: 3 iterations, 21 subsets). The PET images were acquired directly after the voiding of the bladder. The delayed PET imaging of the pelvis was performed 120–180 min post injection (median time: 130 min), with a 4 min acquisition time per bed position and with the same reconstruction protocol. The PET/CT image analysis was performed using the Siemens Workstation (Syngovia, MMWS, Siemens Medical Solutions Inc., Malvern, PA, USA).

The images were analyzed separately by two nuclear medicine physicians. In visual analysis, an abnormal uptake was considered as a positive lesion when it showed a non-physiological increase in uptake that was discernible above background activity. Due to the fact that the cut-off decrease in SUVmax is not strictly defined, we had assumed that when the lesion’s uptake decreased over 30% in the delayed phase, it was classified as a non-malignant lesion [12,19,20]. Since the study construction did not include histopathological analysis of patients, due to the treatment procedure, which had not involved operations of patients, we were not able to compare lesions to the gold-standard pathomorphological study. That is why we were not able to classify lesions as false or true, positive, or negative. For quantitative analysis, the maximal standard uptake value (SUVmax) of a positive lesion was measured on Ga-68-PSMA-11 PET/CT images with spherical volumes of interest (VOIs).

### 2.3. Statistical Methods

Variables were evaluated for distribution pattern. Mean and standard deviations were used to summarize the patients’ characteristics. Statistical analysis was performed on the PQStat (version 1.8.0 b.304, 2020, PQStat Software) with adequate tests (Lilliefors test for normality, Shapiro–Wilk test for normality, D’Agostino–Pearson test for normality, Wilcoxon Mann–Whitney test, Wilcoxon matched-pairs test).

### 2.4. Ethics

This single-institution study was approved by the Ethical Committee of the Medical University of Warsaw (KB/235/2016). Written informed consent was obtained from all patients according to institutional guidelines.

## 3. Results

### 3.1. Patients

#### 3.1.1. Flowchart

To the study we have enrolled 154 patients, 46 patients were excluded from the study due to not matching the inclusion criteria. The flowchart of the study is presented in Figure 1. 

#### 3.1.2. Patients’ Characteristics

The patients’ characteristics are shown in Table 1.

### 3.2. PET/CT Ga-68-PSMA-11 Detection Rate

No adverse events induced by the usage of the diagnostic radiopharmaceutical were reported.

Of the 108 patients, 86 (79.6%) with a mean PSA of 3.6 ± 5.8 ng/mL and a median of 1.8 (range: 0.3–39.0) showed at least one focal uptake suggestive of recurrent prostate cancer on the Ga-68-PSMA-11 PET/CT images. In 22 patients with a mean PSA of 1.8 ± 3.2 ng/mL and median of 0.8 (range: 0.2–16.0), we did not find any lesions in the study after 60 min and in the delayed study. PSA levels were statistically lower in patients with negative Ga-68-PSMA-11 PET/CT results than in patients with positive results (*p* = 0.0065).

From 86 positive patients, 80 patients had foci in the field of view of both phases, and only they were included in the analysis.

Ga-68-PSMA-11 PET/CT detection rate for PSA: 0.2 < PSA < 0.5 ng/mL vs. 0.5 ≤ PSA < 1.0 ng/mL vs. 1.0 ≤ PSA < 2.0 ng/mL vs. PSA ≥ 2.0 ng/mL was 56% (the same for standard and delayed phase) vs. 60% (standard vs. delay: 52 vs. 60%) vs. 87% (standard vs. delay: 83 vs. 87%) vs. 82% (standard vs. delay: 77 vs. 82%) of patients, respectively.

Ga-68-PSMA-11 PET/CT showed positive examination in 75% with Gleason score 5, 61% with Gleason 6, 83% with Gleason 7, 89% with Gleason 8, 82% with Gleason 9, and 100% with Gleason 10.

### 3.3. PET/CT Ga-68-PSMA-11 Lesions-to-Lesion Analysis

We found 284 lesions. To perform the lesion-to-lesion analysis, we had to choose only lesions that were in the field of view of the standard and the delayed study. We found 198 lesions meeting these conditions in 80 patients.

The statistically delayed phase of the study was associated with a higher SUVmax compared to the standard study (*p* = 0.000037). The median SUVmax_60_ was 5.5 (range: 0.9–68.6), while SUVmax_120_ was 7.3 (range: 1.0–72.9)

We separately analyzed lesions in the pelvis classified as local recurrence, lymph nodes, and others.

#### 3.3.1. Local Recurrence

When we analyzed the local recurrence, we found 33 lesions in 28 patients. One patient (3%) had a lesion visible only in the delayed phase image. The other patient had two lesions in the prostate gland, one of them was visible only in the delayed phase image. One patient had two lesions in standard time, whereas after 120 min, only one was found. The rest of the patients with local recurrence had lesions visible in both phases (Table 2). The median SUVmax_60_ was 7.3 (range: 0.9–47.5), and SUVmax_120_ was 8.7 (range: 1.0–58.2) (*p* = 0.00005).

Depending on the Gleason score and PSA value, the difference in the median SUVmax 60 vs. 120 min was statistically significant for Gleason 7 and 8 and PSA value >1 ng/mL (Table 3 and Table 4). The significant differences in SUVmax changes over the time for primary treatment were observed (Table 5).

#### 3.3.2. Local Lymph Node Metastases

When we analyzed the local lymph nodes, we found 122 lesions in 48 patients.

The median SUVmax_60_ was 5.5 (range: 0.9–68.6), and SUVmax_120_ was 7.6 (range: 1.0–72.9) (*p* < 0.000001). Seven (14%) patients had lesions visible only in the delayed phase image.

In the delayed phase, we observed a decrease in SUVmax in three foci, which we interpreted as inflammatory lesions. In the other 119 lymph nodes, an increase in SUVmax was observed.

Depending on the Gleason score and PSA value, the difference in the median SUVmax 60 vs. 120 min was statistically significant in most cases (Table 6 and Table 7).

#### 3.3.3. Other Local Metastases

When we analyzed the other local metastases, we found 44 foci in 32 patients. The other lesions were in the seminal vesicle (6), bone (15), penis (3), rectum (5), and other (15).

The median SUVmax_60_ was 4.9 (range: 0.9–34.6), and SUVmax_120_ was 5.9 (range: 1.0–40.3) (*p* > 0.05).

In the delayed phase, we observed a decrease in SUVmax in 14 foci, which we interpreted as inflammatory/non-specific changes. In the other 30 lesions, an increase in SUVmax was observed.

Depending on the Gleason score, the difference in the median SUVmax 60 vs. 120 min was statistically significant (Table 8 and Table 9). Only PSA values ≥2 ng/mL in the median SUVmax 60 vs. 120 min were statistically significant. However, the number of remaining patients in the group was <4.

### 3.4. Impact on Clinical Management

The delayed phase had a great impact on changing the clinical procedure in 14/86 patients (16%).

In seven (8%) patients, when increased uptake was seen only in the study after 60 min and not visible in the delayed phase, the foci were interpreted as physiological or inflammatory accumulation.

Five of them showed only one focus of abnormal tracer accumulation, which was interpreted as an inflammatory sign. In those cases, the Ga-68-PSMA-11 PET/CT images’ result was no site of PC recurrence. The final decision about the treatment was made by the clinician according to clinical data.

The other two patients, in whom lesions declined in the delayed study, had only local recurrences. In this case, it was possible to qualify them to salvage radiotherapy.

On the contrary, only the delayed images showed increased tracer accumulation in seven (8%) patients. One of them showed local recurrence and was qualified to salvage radiotherapy. In four of them, the Ga-68-PSMA-11 PET/CT showed uptake in regional lymph nodes, which allowed direct experimental treatment planning: radiotherapy or surgery. In the other two patients with local recurrence, the delayed phase showed additional foci, and systemic treatment was indicated.

#### Individual Examples

Ga-68-PSMA-11 PET/CT images of a 69-year-old man with prostate cancer, Gleason score 7 (3 + 4). Thirteen years after RP, the follow-up test showed an increase in the level of PSA. The PSA level was 4.05 ng/mL one day before examination.

The Ga-68-PSMA-11 PET/CT images showed increased tracer accumulation in the right hip plate in the study after 60 min with SUVmax 8.3, without structural lesion in the CT image (arrow in the upper row, CT, PET, and fused PET/CT images). However, in the study after 120 min, there was no increased accumulation of Ga-68-PSMA-11 in the corresponding section of the study (Figure 2 arrow in the lover row, CT, PET, and fused PET/CT images). Due to the washout of the radiotracer, the hot spot was classified as an inflammatory lesion, which changed the diagnosis from local and disseminated recurrence to local recurrence only.

Ga-68-PSMA-11 PET/CT images of a 74-year-old man with PC, Gleason score 7 (3 + 4); 17 months after laparoscopic RP, and 15 months after radiotherapy on the prostate bed. During the follow-up, the patient complained of illness in the right iliac fossa; the PSA level began to rise with a level of 0.51 ng/mL 8 days before the study, with 2 months’ doubling time. No recurrence was found in CT and bone scintigraphy.

Ga-68-PSMA-11 PET/CT images after 60 min were normal; after 120 min only, an internal iliac lymph node 3 mm in size with increased tracer accumulation (SUVmax 5.6) was shown (Figure 3 arrow in the lower row, CT, PET, and fused PET/CT images). Due to the marker accumulation in the delayed study, the result of the study was changed, and the study was interpreted as a recurrence in lymph nodes.

## 4. Discussion

In prostate cancer patients, initially managed with curative intent, in whom BCR is observed, the proper determination of the sites of disease relapse, especially differentiation between local recurrence, nodal involvement, or distant dissemination, appears to be essential. It may serve as a prognostic factor in terms of treatment outcomes, but it can also significantly impact the further clinical decision-making process [21]. The recently updated EAU guideline recommends PSMA PET/CT as the first-choice diagnostic modality in patients with PSA-only recurrence at a PSA level above 0.2 ng/mL. PSMA PET/CT results could have an influence on subsequent management decisions in patients after RP and on post radiotherapy to be considered fit for salvage treatment [22,23]. According to the guidelines, disease relapse limited to the site of primary treatment may be an indication for performing salvage radiotherapy or salvage prostatectomy, but detection of distant metastases would make local salvage treatment unreasonable. At the same time, accurate assessment of metastasis site and volume might be helpful in identifying patients with oligometastatic disease or nodal-only recurrence, in whom several surgical or radiotherapeutic metastasis-directed treatments were proposed, with studies reporting promising results [24,25,26]. The idea of salvage therapy involves irradiation or removal of all metastatic sites of the recurrent disease. Accurate assessment before treatment is of key importance. It has been proven that whole-pelvis irradiation results in longer BCR-free survival [27]. However, the standard salvage radiotherapy targets only prostate fossa with or without the base of seminal vesicles [28]. The current study suggests that the delayed phase in the PSMA PET/CT scan may indicate the irradiation area better. However, the oncological effect of this change is uncertain. Ongoing trials OLIGOPELVIS (NCT02274779) and PEACE V (NCT03569241) aim to find the proper radiotherapeutic template in different stages of nodal recurrence [29,30].

Moreover, nowadays, there is a growing interest in the surgical treatment of nodal recurrence. Plousard et al. demonstrated that PSMA PET/CT has been used in most modern series of salvage lymph node dissections [31]. The delayed phase of PSMA PET/CT, with a better detection rate, may suggest a more extended template of salvage surgery, leading to better oncological results. It can be strongly connected with radioguidance, which is becoming more popular with connection to the surgical treatment of nodal recurrence. Usage of a gamma probe and PSMA conjugated with a radioisotope may show foci of suspected metastasis during the operation [32,33]. Due to the different duration of the operation, the anesthesia procedure, obtaining the appropriate surgical field and access to the locations, confirmation of the uptake of PSMA, and even the increase in tracer accumulation over a longer period of time may support that idea.

The EUA Guidelines 2022 also mention that not only PSMA PET/CT plays a role in the diagnosis of suspicious lymph nodes in prostate cancer. It is worth mentioning the increasing interest in using indocyanine green (ICG) in primary surgical treatment. Intraprostatic injection of ICG improves intraoperative identification of prostatic lymph nodes, resulting in a higher number of dissected lymph nodes and retrieved lymph node metastases. However, the impact of biochemical recurrence on those patients is still unknown [34,35]

There is emerging evidence that PSMA PET/CT has excellent performance in detecting recurrent lesions and metastatic lymph nodes in prostate cancer, which has an impact on the treatment plans of BCR patients. The acquisition procedure plays an important role in molecular imaging. For PSMA PET/CT studies, several acquisition protocols with different acquisition times have been proposed. The standard procedure for PSMA PET/CT imaging recommends a 60 min interval uptake time [11,18], but others suggest early dynamic and delayed phase 3–4 h p.i. imaging [13,14,16,17]. However, the 3–4 h phase, especially in late scanning patients, is difficult to perform in busy clinical environments. On the other hand, the dual-phase protocol has been used in several tracers and indications, e.g., in lung cancer, F-18-FDG PET has been used to improve the diagnostic efficacy and to differentiate benign from malignant lesions [36]. Another important aspect that must be considered here is the 68 min half-life of Ga-68, which has a significant impact on the statistical count in the late delayed phase.

The present study evaluated the advantages of Ga-68-PSMA-11 in 108 patients with BCR of prostate cancer after primary treatment with curative intent at 60 and 120–180 min after the injection of radiopharmaceutical. The strongest point of the presented study is its prospective nature in comparison to previously published retrospective data. The overall detection rate in our study was 79.6%, which is comparable to the published data. Only one study reports a different detection rate (49.6%); however, it only included patients with a PSA below 1 ng/mL in the inclusion criteria [37].

The main important finding, from a clinical point of view, is the change to the clinical management in 16% of examined patients. If we look at the impact of the delayed examination, we can say that it allows detecting unusually small lesions and differentiating between physiological and inflammatory lesions.

In our study, 8% of patients had increased uptake only in the 60 min study, which was interpreted as physiological or inflammatory accumulation. It allowed the application of salvage radiotherapy in two patients with local recurrence only. In the rest of the patients, the final decision about the treatment was taken by the clinician according to clinical data.

Furthermore, in 8% of patients, only delayed images showed increased tracer accumulation. Thus, Ga-68-PSMA-11 allowed targeted radiotherapy planning in five patients: four patients with only regional lymph nodes recurrence and one patient with local recurrence, and due to disseminated disease in the other two patients, systemic treatment was started.

However, it is still debated whether targeted radiotherapy at lymph node metastases should be considered as a means of therapy despite its efficacy or if systemic treatment should be started. In our cases, Ga-68-PSMA-11 was used for experimental targeted therapy planning with good clinical and biochemical effects.

In currently published studies, there are limited data about the impact of the delayed phase of PSMA PET/CT in treatment management. Most studies focus on the SUV changes between standard and delayed phases, as well as on improving tumor visualization due to lower background activity.

Hoffmann et al. and Derlin et al. found that delayed imaging did not increase the number of detected metastases significantly. They did not observe the influence of additional late imaging on therapeutic decisions [38,39]. Considering individual cases, in a few of them, additional delayed scans provided information advantageous in lesion detection, whereas delayed imaging Ga-68-PSMA detected suspicions of malignancy exclusively in an additional 11 (5%) lesions. A total of 33 lesions suspected of being malignant in the standard phase were confirmed as malignant by increased tracer uptake in the delayed scans [38]. Beheshti et al. observed exclusive detection in delayed images of Ga-68-PSMA I&T accumulation in 10/240 (4%) additional patients; 9 of them had local recurrences, underlining that these may be particularly difficult to distinguish in non-diuretic standard images [37]. Schmuck et al., in delayed Ga-68-PSMA I&T PET/CT imaging at 3 h p.i., exclusively identified pathologic findings in 5.4% (10/184) of patients [16,40].

If we focus on the impact of the delayed phase on SUV value or tumor to background changes, most published data strongly support the delayed phase. Firstly, as shown in early pharmacokinetic studies, background activity decreases significantly with time, between 1 and 3 h p.i., resulting in an improved tumor/background (T/B) ratio [41], and the same has been emphasized by other authors [16,40,42]. In our study, we observed the same impact on local recurrence, lymph node involvement, and other local metastases in the majority of cases, with significant differences in SUVmax_60_ and SUVmax_120._

The novel tracer based on F-18, which has a nearly 2 times longer half-life than Ga-68, should be favorable in the case of delayed studies. Authors exploring F-18-JK-PSMA-7 showed that SUVmax and SUVpeak increased by up to 60 min p.i., remained at this intensity in the subsequent PET/CT scans up to 140 min p.i, and then, after, in the period between 170 and 200, a further significant increase was observed. They concluded that a late acquisition window, even up to 3 h after injection, may be favorable for this F-18-JK-PSMA-7 [43]. For other tracers, such as F-18-DCFPyL and F-18-PSMA-1007, images at 120 min after injection presented a higher detection rate of prostate cancer characteristic lesions than images at 60 min after injection. A recently published paper with F-18-DCFPyL confirmed that dual-phase imaging is useful in the differential diagnosis of malignant lesions and benign lesions [44].

There are some limitations to this study. The first one is the lack of histopathological verification of the found foci. Thus, we cannot show false-positive lesions. Although it could be considered as a weakness, in clinical practice, it is not possible to perform it in the majority of cases. In patients after RP, when Ga-68-PSMA-11 shows a 4 mm lymph node, it is almost impossible to find it without radioguided surgery. Access to technetium-based radiopharmaceuticals is still very limited. Obtaining the histopathological results would allow us to calculate the sensitivity and specificity of both standard and delayed phases.

Secondly, the study protocol was designed to evaluate the impact of a delayed image on clinical management, but not on the interobserver agreement. The final result was a consensus between two readers, both of them taking into consideration changes in the SUVmax in standard and delayed phase together. There has been no separately analyzed standard and delayed phase images. The data of the individual readers’ results were not collected, so the Kappa value cannot be evaluated.

Lastly, we evaluated only one time point in the delayed phase. However, as we have data in the literature on delayed phase 3–4 h pi, due to practical reasons, we tested 2–3 h pi.

## 5. Conclusions

The Ga-68-PSMA-11 PET/CT delayed phase examination has an impact on the change in patient management by clarifying equivocal findings and providing a better chance for therapy planning. The late images changed clinical management in 16% of patients.

Most detected lesions showed a significant increase in tracer uptake at the delayed phase with improved contrast compared to the standard imaging.

In many situations, the delayed phase cannot be applied to all patients because of the practical considerations of a busy clinical environment. However, it should be considered as a priority for patients with BCR and negative standard 60 min post injection examination, or in cases where it can provide additional information on the number and location of metastatic lesions (to select the most effective treatment).

## Figures and Tables

**Figure 1 jcm-11-03311-f001:**
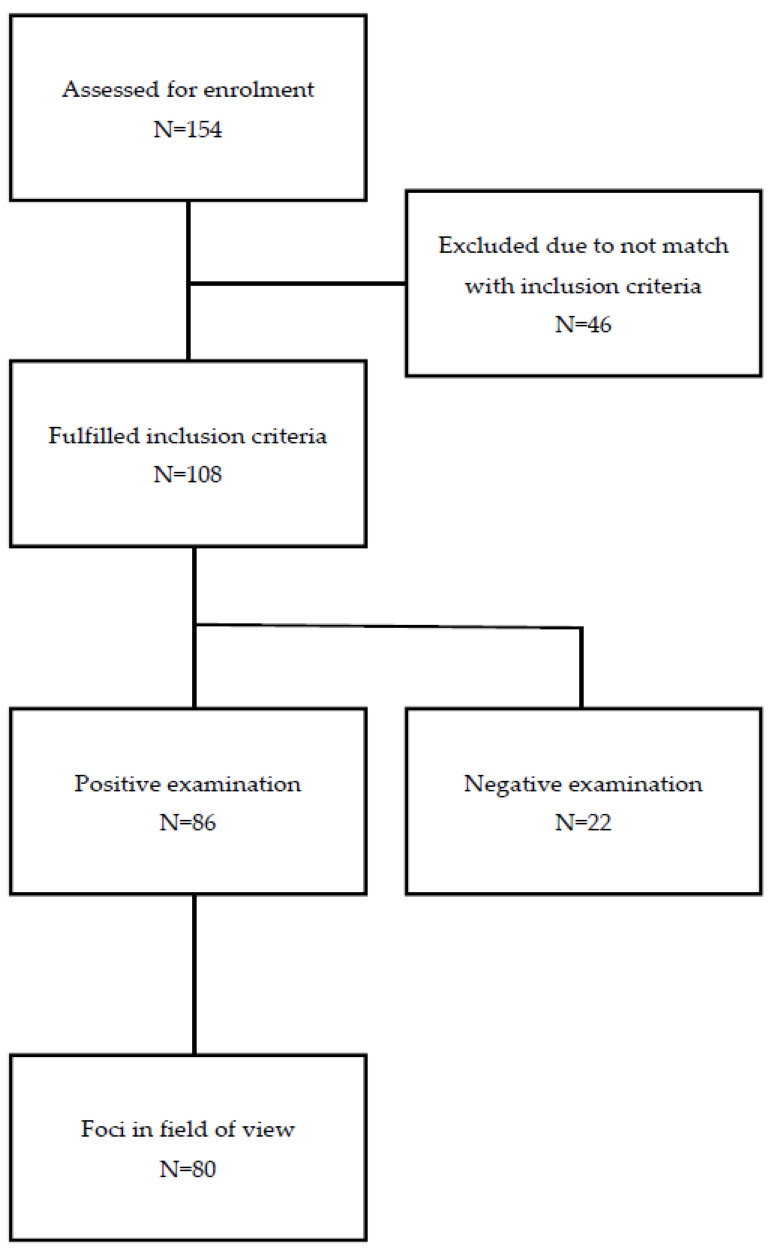
The flowchart of the study.

**Figure 2 jcm-11-03311-f002:**
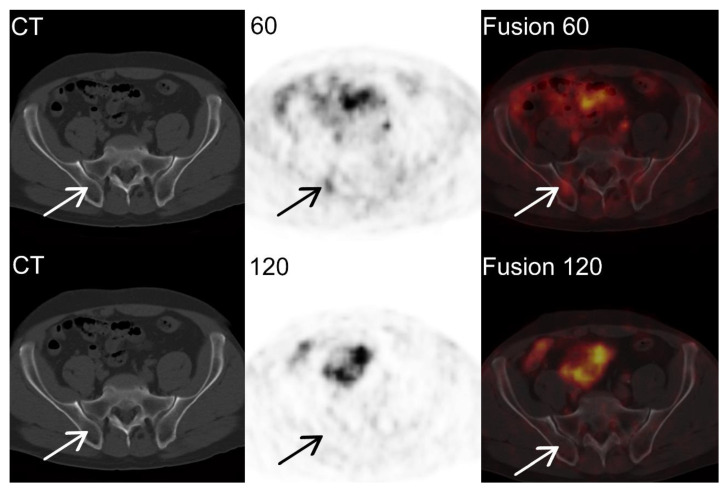
Ga-68-PSMA-11 PET/CT images of false positive 60 min examination.

**Figure 3 jcm-11-03311-f003:**
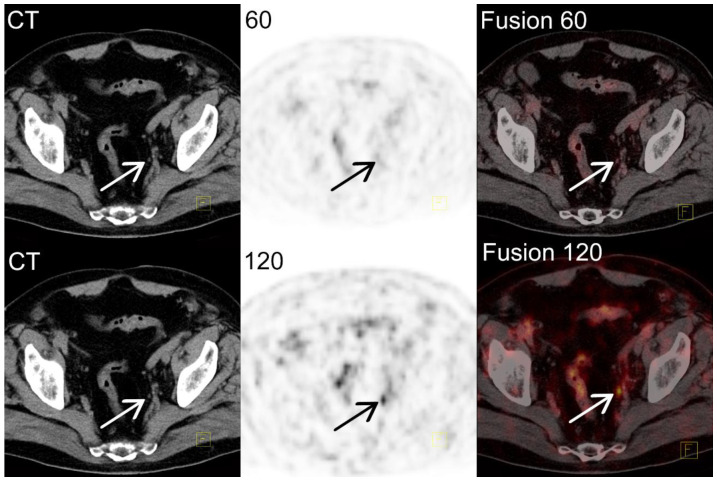
Ga-68-PSMA-11 PET/CT images of false negative 60 min examination.

**Table 1 jcm-11-03311-t001:** Patients’ characteristics.

Characteristics	Parameters (%)
Number of patients	108
Age (y)	
Median	68.5
Range	49–83
Primary Gleason score	
5	4 (4%)
6	13 (12%)
7	54 (50%)
8	18 (17%)
9	17 (16%)
10	2 (2%)
Treatment	
Surgery	40 (37%)
Surgery + RTx	44 (41%)
RTx	22 (20%)
HIFU	2 (2%)

**Table 2 jcm-11-03311-t002:** Local recurrence 60 min vs. 120 min of Ga-68-PSMA-11 PET/CT.

	Local Recurrence (Per-Patient Detection Rate)	Number of Lesions (Lesion-Based Detection Rate)	False Positive (Per-Patient Detection Rate)
60 min	27	31	1
120 min	28	32	0
60 + 120	28	33	1

**Table 3 jcm-11-03311-t003:** Depending on Gleason score, the overall uptake in median SUVmax of local recurrence 60 min vs. 120 min of Ga-68-PSMA-11 PET/CT (lesion-based analysis).

	Gleason 5 (n = 4)	Gleason 6 (n = 4)	Gleason 7 (n = 13)	Gleason 8 (n = 9)	Gleason 9 (n = 3)
SUVmax_60_	6.3	5.5	6.4	9.1	7.9
SUVmax_120_	5.8	8.6	7.2	13.5	10.6
*p*	ns	ns	0.0024	0.0092	ns

**Table 4 jcm-11-03311-t004:** Depending on PSA value, the overall uptake in median SUVmax of local recurrence 60 min vs. 120 min of Ga-68-PSMA-11 PET/CT (lesion-based analysis).

	PSA > 0.2 to <0.5 ng/mL (n = 2)	PSA 0.5 to <1 ng/mL (n = 3)	PSA 1.0 to <2.0 ng/mL (n = 8)	PSA ≥ 2 ng/mL (n = 20)
SUVmax_60_	5.1	3.2	8.0	7.3
SUVmax_120_	8.0	5.8	10.2	8.7
*p*	ns	ns	0.042	0.0038

**Table 5 jcm-11-03311-t005:** Depending on primary radical treatment, the overall uptake in median SUVmax of local recurrence 60 min vs. 120 min of Ga-68-PSMA-11 PET/CT (lesion-based analysis).

	Prostatectomy (n = 12)	RTx (n = 17)	Prostatectomy + RTx (n = 4)
SUVmax_60_	7.8	6.3	7.6
SUVmax_120_	8.6	8.6	9.6
*p*	0.0096	0.0075	ns

**Table 6 jcm-11-03311-t006:** Depending on Gleason score, the overall uptake in median SUVmax of lymph nodes in 60 min vs. 120 min of Ga-68-PSMA-11 PET/CT (lesion-based analysis).

	Gleason 6 (n = 4)	Gleason 7 (n = 71)	Gleason 8 (n = 25)	Gleason 9 (n = 16)	Gleason 10 (n = 3)
SUVmax_60_	13.9	5.5	5.6	3.8	4.8
SUVmax_120_	22.0	7.5	13.8	5.4	3.9
*p*	ns	<0.000001	0.00002	0.00085	ns

**Table 7 jcm-11-03311-t007:** Depending on PSA value, the overall uptake in median SUVmax of lymph node in 60 min vs. 120 min of Ga-68-PSMA-11 PET/CT (lesion-based analysis).

	PSA > 0.2 to <0.5 ng/mL (n = 6)	PSA 0.5 to <1 ng/mL (n = 16)	PSA 1.0 to <2.0 ng/mL (n = 23)	PSA ≥ 2 ng/mL (n = 74)
SUVmax_60_	4.7	5.0	3.9	6.0
SUVmax_120_	7.8	6.4	6.7	9.5
*p*	ns	0.0015	0.00017	<0.000001

**Table 8 jcm-11-03311-t008:** Depending on Gleason score, the overall uptake in median SUVmax of other local metastases in 60 min vs. 120 min of Ga-68-PSMA-11 PET/CT (lesion-based analysis).

	Gleason 6 (n = 8)	Gleason 7 (n = 12)	Gleason 8 (n = 7)	Gleason 9 (n = 3)
SUVmax_60_	7.2	5.7	12.9	4.8
SUVmax_120_	8.0	6.8	17.1	10.9
*p*	ns	0.037	0.022	ns

**Table 9 jcm-11-03311-t009:** Depending on PSA value, the overall uptake in median SUVmax of other local metastases in 60 min vs. 120 min of Ga-68-PSMA-11 PET/CT (lesion-based analysis).

	PSA > 0.2 to <0.5 ng/mL (n = 2)	PSA 0.5 to <1 ng/mL (n = 4)	PSA 1.0 to <2.0 ng/mL (n = 3)	PSA ≥ 2 ng/mL (n = 21)
SUVmax_60_	4.9	2.8	12.8	7.4
SUVmax_120_	6.5	5.9	18.2	9.7
*p*	ns	ns	ns	0.0038

## Data Availability

Not applicable.

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
