# Peer review of "Ga-68-PSMA-11 PET/CT in Patients with Biochemical Recurrence of Prostate Cancer after Primary Treatment with Curative Intent—Impact of Delayed Imaging"

_jcm, 2022, doi:10.3390/jcm11123311_

Round 1

Reviewer 1 Report

Dear Authors,

article focus on an interesting topic, perhaps some points should be further discussed.

1)Abstract: it would be nice to add a comparison between patients who had a significant impact from delayed phase vs the others, with a p value

2)Materials and methods: inclusion and exclusion criteria should not be a bulleted lists, please rewrite it

3) Materials and methods: in M&Ms you reported also the guidelines on how to write it. Please delete them

4) Materials and methods: patients' characteristics and Table 1 should be moved in results, as they are results, together with flowchart with patients numbers

5) Materials and methods: ethics should have its own paragraph

6) Results: tables alike table 2 for metastases and lymph nodes should be welcome, with comparisons between 60 and delayed phase detection rate. 

7) Discussion: some phrases seems not linked one to another, thus it should be improved style, fluency and cohesion of the discussion itself

8) Abbreviations, when applied, should be used throughout the manuscript, thus if you apply RP for radical prostatectomy, it should not be found radical prostatectomy in the subsequent part of the manuscript

Author Response

Dear Authors,

article focus on an interesting topic, perhaps some points should be further discussed.

1)Abstract: it would be nice to add a comparison between patients who had a significant impact from delayed phase vs the others, with a p value

Thank you for this remark, we added the p value as suggested. The aim of this study was a clinical impact on the patient’s management. It is clearer to see in how many patients the delay phase study can change the previous result. That’s results are described in the text. 

2)Materials and methods: inclusion and exclusion criteria should not be a bulleted lists, please rewrite it

Thank you for this remark, we edited as suggested.

3) Materials and methods: in M&Ms you reported also the guidelines on how to write it. Please delete them

Thank you for this remark, we edited as suggested.

4) Materials and methods: patients' characteristics and Table 1 should be moved in results, as they are results, together with flowchart with patients numbers

Thank you for this remark, we edited as suggested with additional 3.1.1 and 3.1.2 paragraph

5) Materials and methods: ethics should have its own paragraph

Thank you for this remark, we edited as suggested.

6) Results: tables alike table 2 for metastases and lymph nodes should be welcome, with comparisons between 60 and delayed phase detection rate. 

Thank you for this remark, we had tried to prepare the table as suggested but due to the differences in lymph node numbers in both phases it is less visible that the description in the text.

7) Discussion: some phrases seems not linked one to another, thus it should be improved style, fluency and cohesion of the discussion itself

Thank you for this remark, we edited as suggested.

8) Abbreviations, when applied, should be used throughout the manuscript, thus if you apply RP for radical prostatectomy, it should not be found radical prostatectomy in the subsequent part of the manuscript

Thank you for this remark, we edited as suggested.

Reviewer 2 Report

The authors provide a nice manuscript detailing the difference in patient management after the implementation of delayed images with PSMA PET/CT. However, there could be a few improvements for the manuscript to make it more robust.

The main concepts of the paper is clear - delayed images at 120 mins provide more information regarding the PSMA uptake. The most commonly used radiotracers in the USA (per package inserts) recommend 60 minutes for F18-DCFPyL and 50-100 mins for Ga-68 PSMA-11.

- The authors could include percentages in table 1 to make it more relevant to the readers

  • There are copy/pasted sections in the lines 146-160 and 403-406 which should be edited.
  • The tables with SUVmax should include if there are mean or median SUVmax from the particular patient groups (the body of the manuscripts says median)
  • While the impact of delay phase is significant, the importance of this finding should be discussed in the discussion session and not in the findings (line 237)

The examples the authors have shown are good, however if the figure description is being submitted as an underlying explanation to the Figures 1 and 2, they can be more concise to reflect the only important factors (i.e. 69-year-old male with Gleason score 7 with PSA of 4.05, suspicious right sacraoiliac uptake which disappears in the delay phase) and the rest of the explanation of the washout in the tracer can be discussed in the discussion session. Or if the authors could include a separate section for individual examples (or as an appendix) in the form of part 3.3.1 in the findings. 

The main discussion regarding this paper is however the actual correlation - meaning there is no factual data when it comes to a PSMA avid lesion is considered false positive if the uptake decreases in delay images or vice versa. It is very common to proceed with biopsies or in some instances with surgery after reporting these metastases. The question for the authors if whether they have any sort of follow up after the scan via chart review, which would make this study more robust. This can be in form of histopathological analysis (despite the study says it might be hard to biopsy 4 mm nodes) or in the way of referring patients to the additional therapy. I would love to see how many of the referring physicians acted based upon the delayed image findings. 

As I stated above, many centers perform PSMA scans at 60 or 90 minute mark. It is not feasible to scan patients twice in busy centers and the current study could compare if there is any data regarding the differences in 90 minute imaging vs 120 min imaging. If the results are similar to 120 min imaging there is no point of scanning patients twice. 

Finally the manuscript can be made more concise by combining some paragraphs in the discussion and after careful re-read with some attention to abbreviations (i.e. HIFU). Some changes should be made in the abstract to make it grammatically correct. 

Author Response

The authors provide a nice manuscript detailing the difference in patient management after the implementation of delayed images with PSMA PET/CT. However, there could be a few improvements for the manuscript to make it more robust.

The main concepts of the paper is clear - delayed images at 120 mins provide more information regarding the PSMA uptake. The most commonly used radiotracers in the USA (per package inserts) recommend 60 minutes for F18-DCFPyL and 50-100 mins for Ga-68 PSMA-11.

- The authors could include percentages in table 1 to make it more relevant to the readers

Thank you for this remark, we edited as suggested.

  • There are copy/pasted sections in the lines 146-160 and 403-406 which should be edited.

Thank you for this remark, we edited as suggested.

  • The tables with SUVmax should include if there are mean or median SUVmax from the particular patient groups (the body of the manuscripts says median)

Thank you for this remark, we edited tables tittles as suggested.

  • While the impact of delay phase is significant, the importance of this finding should be discussed in the discussion session and not in the findings (line 237)

Thank you for this remark, we analyse this impact in discussion line 331-343. We left it in finding as a summary to clarify clinical impact.

The examples the authors have shown are good, however if the figure description is being submitted as an underlying explanation to the Figures 1 and 2, they can be more concise to reflect the only important factors (i.e. 69-year-old male with Gleason score 7 with PSA of 4.05, suspicious right sacraoiliac uptake which disappears in the delay phase) and the rest of the explanation of the washout in the tracer can be discussed in the discussion session. Or if the authors could include a separate section for individual examples (or as an appendix) in the form of part 3.3.1 in the findings. 

Thank you for this remark, we edited as suggested with part 3.4.1 – Individual examples.

The main discussion regarding this paper is however the actual correlation - meaning there is no factual data when it comes to a PSMA avid lesion is considered false positive if the uptake decreases in delay images or vice versa. It is very common to proceed with biopsies or in some instances with surgery after reporting these metastases. The question for the authors if whether they have any sort of follow up after the scan via chart review, which would make this study more robust. This can be in form of histopathological analysis (despite the study says it might be hard to biopsy 4 mm nodes) or in the way of referring patients to the additional therapy. I would love to see how many of the referring physicians acted based upon the delayed image findings. 

Thank you for this remark. This project was focus on impact on PSMA PET/CT delay phase impact. Due to the already established histopathological diagnosis and the confirmed biochemical recurrence, unforunately it is not practice to do biopsy of visible on CT/MR/ PET lesions. The final decision about treatment is made by urologist/oncologist based on the imaging tests together with the increase of markers. Our experience shows that histopathological confirmation of recurrence by biopsy or surgery is performed very rarely, only in situations that raise many doubts. In this study, we do not have information about the conducted tests confirming the diagnosis. We understand the importance of a biopsy and its impact on the final result of the study, therefore it is one of the greatest limitations of this study. It is a very good idea for a future study, the complexity of which will be much greater, which will translate into a greater burden on patients and cooperation only with selected treatment centers.

As I stated above, many centers perform PSMA scans at 60 or 90 minute mark. It is not feasible to scan patients twice in busy centers and the current study could compare if there is any data regarding the differences in 90 minute imaging vs 120 min imaging. If the results are similar to 120 min imaging there is no point of scanning patients twice. 

Thank you for this remark, according to our knowledge and study results, we have no data about differences in 90minutes imaging vs 120 min imaging. We made analysis based on our results, the organization of work in our department allows us to do the first scan after 60 minutes, but for us it would be very difficult to do it after 90 minutes or to do the additional imaging after 90 minutes.  

Finally the manuscript can be made more concise by combining some paragraphs in the discussion and after careful re-read with some attention to abbreviations (i.e. HIFU). Some changes should be made in the abstract to make it grammatically correct. 

Thank you for this remark. We reread the text and made corrections.

Reviewer 3 Report

Overall interesting results, congratulations.

One minor comment before acceptance. Any information about the yield of PLND among patients who underwent radical prostatectomy? Considering the number of local recurrences these data are worth mentioning. As the authors correctly stated that "surgical treatment of nodal recurrence leads to a growing interest in radioguidance. Usage of gamma probe and the PSMA conjugated with radioisotope may show foci of suspected metastasis during the operation". Here, I suggest to further refer to this work (doi: 10.1097/RLU.0000000000002600). In this specific setting Indocyanine Green (ICG) guidance is also worth mentioning as a cost effective, reliable tool to further improve the extension of primary PLND. Particularly, ICG guidance may lead to an increase yield of targeted PLND associated with a better BCR-free survival rate as pointed out in a recent work  (doi: doi: 10.1111/iju.14513). The author should consider and discuss these findings.

Author Response

Overall interesting results, congratulations.

One minor comment before acceptance. Any information about the yield of PLND among patients who underwent radical prostatectomy? Considering the number of local recurrences these data are worth mentioning.

Thank you for this remark, unfortunately we had no access to the full detailed histopathological data. This is very interesting idea, but needs to plan in advance a very good cooperation between the urologist and nuclear medicine department.

As the authors correctly stated that "surgical treatment of nodal recurrence leads to a growing interest in radioguidance. Usage of gamma probe and the PSMA conjugated with radioisotope may show foci of suspected metastasis during the operation". Here, I suggest to further refer to this work (doi: 10.1097/RLU.0000000000002600). In this specific setting Indocyanine Green (ICG) guidance is also worth mentioning as a cost effective, reliable tool to further improve the extension of primary PLND. Particularly, ICG guidance may lead to an increase yield of targeted PLND associated with a better BCR-free survival rate as pointed out in a recent work  (doi: doi: 10.1111/iju.14513). The author should consider and discuss these findings.

Thank you for this remark, we added citation and additional paragraph:

EUA Guidelines 2022 also mention that not only PET PSMA has the role in diagnosis suspicions lymph nodes in prostate cancer. Worth mentioning is increasing interest of using indocyanine green (ICG) in primary surgical treatment. Intraprostatic injection of ICG improve intraoperative identification of prostatic lymph node, resulting in a higher number of dissected lymph nodes and retrieved lymph node metastases. But the impact for biochemical recurrence on those patients is still unknown. [34-35]

Round 2

Reviewer 1 Report

Dear Authors, 

thank you for the punctual response to every point previously risen. There are no further points to be addressed, thus article might be considered for publication in my opinion

Author Response

Dear Authors, 

thank you for the punctual response to every point previously risen. There are no further points to be addressed, thus article might be considered for publication in my opinion

Thank you for all comments, thanks to them the article has gained in quality and readability.

Reviewer 2 Report

Minor spell checks are still required. 

Author Response

Minor spell checks are still required. 

Thanks for your comments, we have made corrections.